# A Generalizable and Efficient Symbolic Regression Method for Time Series Analysis

## Abstract

Current time series analysis methods predominantly rely on quantitative approaches, providing accurate yet often superficial statistical indicators. However, these methods struggle to capture the underlying evolution patterns and lack intuitive, qualitative insights. This paper addresses these gaps by seeking explicit mathematical expressions for the time-varying nature of time series, offering a more intuitive understanding. We frame this task as a combinatorial optimization problem and propose a reinforcement learning-inspired approach. Using Monte-Carlo Tree Search (MCTS) as the basis, we incorporate symbolic regression to derive expressions for the non-linear dynamics in time series. To overcome the inefficiencies and excessive randomness in MCTS, we enhance it with neural networks, forming the **N**eural-**E**nhanced **Mo**nte-Carlo **T**ree **S**earch (NEMoTS) method. This integration leverages neural networks' superior fitting capabilities to introduce priors and replace the simulation phase, significantly improving generalizability and computational efficiency. Experiments on six real-world datasets demonstrate NEMoTS's clear advantages in performance, efficiency, reliability, and interpretability.

## 1 Introduction

Currently, time series analysis frameworks predominantly rely on quantitative tools such as spectral analysis Koopmans (1995); Warner (1998), time-domain analysis Jones (2019); Bence (1995), and moment analysis Brillinger (2002); Gabr (1988). While effective in capturing statistical properties, these methods often fail to provide intuitive and qualitative insights into the underlying mechanisms of time series data. They focus primarily on **how** data evolves over time, but largely overlook **what** drives these changes and **why** specific patterns emerge. By deriving explicit mathematical expressions for time series, we can uncover global evolution patterns across different timeframes Angelis et al. (2023); Makke & Chawla (2022b;a).

For instance, Fig. 1 (A) offers limited information, showing only increasing values and periodic oscillations with growing amplitude, due to the absence of an explicit expression. In contrast, Fig. 1 (B) presents the expression: $f(t) = 0.0974t \left( \log(1.6042t)^{2.65} \right) + 0.9t \cos \left( (0.11t)^{1.66} \right)$. which reveals a logarithmic trend alongside a seasonal component characterized by significant cyclical fluctuations driven by the cosine function. As $t$ increases, both the amplitude and frequency of these fluctuations grow linearly. This example clearly demonstrates how an analytical expression can provide deeper insights in an intuitive and qualitative manner. Particularly for experts, such expressions can be effectively combined with traditional quantitative tools to enhance interpretability and offer a more comprehensive understanding of time series dynamics.

Symbolic regression, a classical and highly interpretable machine learning approach, effectively connects inputs and outputs using mathematical expressions made of basic functions, as highlighted in Makke & Chawla (2022a;b). Symbolic regression uses explicit analytical expressions in a data-driven way to skillfully reveal nonlinear system dynamics without prior constraints (*e.g.*, linear assumptions, polynomial assumptions, or trigonometric function assumptions) Carleson & Gamelin (2013). In time series analysis, this technique not only provides qualitative insights but also enables an in-depth quantitative examination of fundamental evolutionary processes Angelis et al. (2023); Makke & Chawla (2022b). Unlike traditional quantitative methods, symbolic regression delves deeper into the intrinsic dynamics of time series itself, offering substantial insights into the **what**

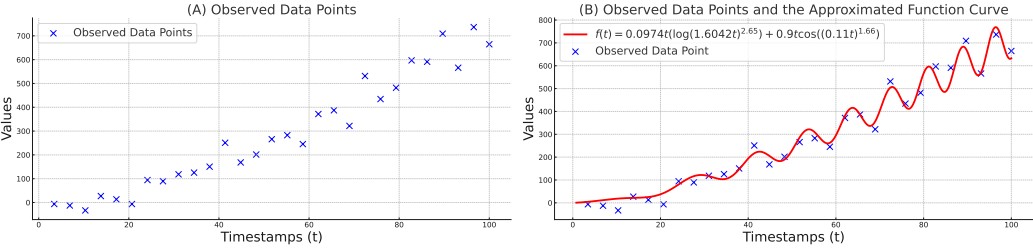

Figure 1: (A) Blue crosses represent the observed values, which is hard to summarize its pattern; and (B) red curve represents the fitted analytical expression of the time series data, which is more intuitive and qualitative for analysis.

and **why** behind evolution. This method excels in analyzing complex, nonlinear systems where standard modeling techniques might fail to grasp system intricacies.

However, current symbolic regression techniques, mainly designed for fitting specific sample, and based on combinatorial optimization methods, often depend on complex heuristic designs to fit a particular case. They use simulation or search algorithms to generate expressions matching that case Angelis et al. (2023); Makke & Chawla (2022b). These methods face challenges such as computational inefficiency, high complexity, and restricted generalization abilities, especially when handling larger datasets Udrescu & Tegmark (2020); Nicolau & Agapitos (2021); Chen et al. (2016). The increased computational requirements and extended model search durations, particularly in extensive iterative processes, diminish their effectiveness in big data scenarios. Additionally, because these techniques concentrate on fitting specific samples, they struggle to identify common patterns across different samples and lack broader learning abilities. This not only hinders performance improvement but also underutilizes the rich knowledge embedded in extensive datasets.

To overcome these limitations, we propose **N**eural-**E**nhanced **Mo**nte-Carlo **T**ree **T**earch (NEMoTS) for time series analysis. NEMoTS uses the Monte-Carlo Tree Search (MCTS) framework, where expressions are represented as tree structures. This approach aligns with parse tree structures and follows context-free grammar rules, ensuring the validity of generated expressions Hopcroft et al. (2001); Kusner et al. (2017). By balancing exploration and exploitation, NEMoTS narrows the search space, improving search efficiency and expression quality compared to other methods Sun et al. (2022); Kamienny et al. (2023). Considering that several challenges still affect the performance of MCTS in symbolic regression, for example: during the selection phase, MCTS relies on random selection without the guidance of a prior distribution, leading to exponential growth of the search space and limited generalization capability. In the simulation phase, the complex rollout operations are computationally demanding and time-consuming. To address these issues, NEMoTS integrates neural networks into the MCTS framework. Neural networks guide the selection of promising nodes, focusing the search, and replace complex simulations with advanced fitting capabilities, streamlining MCTS operations and improving efficiency. This integration also allows NEMoTS to learn from larger datasets, enhancing both fitting accuracy and generalization capacity.

NEMoTS consists of four main components: a pre-defined basic function library, Monte-Carlo Tree Search (MCTS), a policy-value network, and a coefficient optimizer. At its core, MCTS guides the process. In MCTS, the selection and simulation phases are influenced by the policy-value network's assessment and output regarding the overall state of the expression. Each operation within MCTS and the resulting expression originate from a pre-defined basic function library. MCTS produces an initial expression 'backbone', which lacks numerical coefficients. These are then refined by the coefficient optimizer to create a full expression. Expanding on ideas from Sun et al. (2022), we integrate a Symbolic Augmentation Strategy (SAS) during training. SAS improves the simulation phase of the Monte-Carlo tree search by accumulating high-rewarded composite functions. This approach is akin to frequent pattern mining Agrawal et al. (1993), involving the random amalgamation of various basic functions to identify frequent, high-rewarded composite function patterns. These frequently occurring composite functions are then added to the function library based on their average rewards, significantly enhancing the model's fitting abilities.

We carried out comprehensive experiments on six real-world time series datasets. The outcomes reveal that NEMoTS not only excels in symbolic regression tasks for time series, exhibiting exceptional fitting ability and efficiency, but also demonstrates superior performance in extrapolation with the expressions it derives, which implies the reliability of the expressions.

The key contributions of this paper are outlined as follows:

- We utilize symbolic regression to enhance the analysis and understanding of time series data, especially in qualitative aspect. The integration of Monte-Carlo Tree Search (MCTS) in symbolic regression for time series leads to the discovery of high-quality, valid expressions, providing new insights into time series analysis.

- To overcome the inefficiencies and generalization limits of traditional MCTS in time series data, neural networks have been incorporated into the framework. This advancement not only increases the model's efficiency but also expands its learning and generalization capabilities, resulting in enhanced performance.

- Building upon these innovations, we present the **N**eural-**E**nhanced **Mo**nte-Carlo **T**ree **T**earch (NEMoTS), specifically tailored for symbolic regression in time series. The unique inclusion of a symbolic augmentation strategy, inspired by frequent pattern mining, further boosts the model's performance.

- Comprehensive experiments on six real-world time series datasets, NEMoTS demonstrates its remarkable ability and efficiency in symbolic regression tasks for time series.

## 2 PROBLEM DEFINITION

We formally define our task, similar to the classical Empirical Risk Minimization (ERM) approach Vapnik (1991).

**Input:** Given a time series $\mathcal{D} = (t_i, v_i)_{i=0}^{N-1}$ containing $N$ records, where $t_i \in \mathbb{R}$ represents the timestamp and $v_i \in \mathbb{R}$ represents the value corresponding to the timestamp $t_i$.

**Objective:** The goal is to discover an analytical expression $f(\cdot)$ and evaluate it using the following reward function:

$$\mathcal{R} = \frac{\eta^s}{1 + \sum_{i=0}^{N-1} \sqrt{(v_i - f(t_i))^2}}, \tag{1}$$

where $\eta$ is a constant slightly less than 1, and $s$ denotes the size of the generated analytical expression. Generally, the value of this reward function ranges between 0 and 1, balancing the complexity of the generated expression and its fitting degree. The closer it is to 1, the simpler the discovered expression and the higher the achieved fitting accuracy.

## 3 NEURAL-ENHANCED MCTS

### 3.1 MODEL OVERVIEW

The overview of our proposed NEMoTS (**N**eural-**E**nhanced **Mo**nte-Carlo **T**ree **S**earch) for time series is illustrated in Fig. 2.

The NEMoTS comprises four main components: a pre-defined basic function library, Monte-Carlo Tree Search (MCTS), a policy-value network, and a coefficient optimizer. The first three components collaborate to form the basic structure of an expression, named "backbone", which lacks any numerical coefficients. This basic structure is then refined by the coefficient optimizer, which determines appropriate coefficients, resulting in a full expression.

Below, we first outline the collaboration among components:

- **Monte-Carlo Tree Search (MCTS)**: A key component of NEMoTS, MCTS is a four-phase process: selection, expansion, simulation, and back-propagation, with a function library and policy-value network playing vital roles. It creates an expression's structural "backbone," determining layout and basic functions, but not the numerical coefficients, which are set later by a coefficient optimizer.

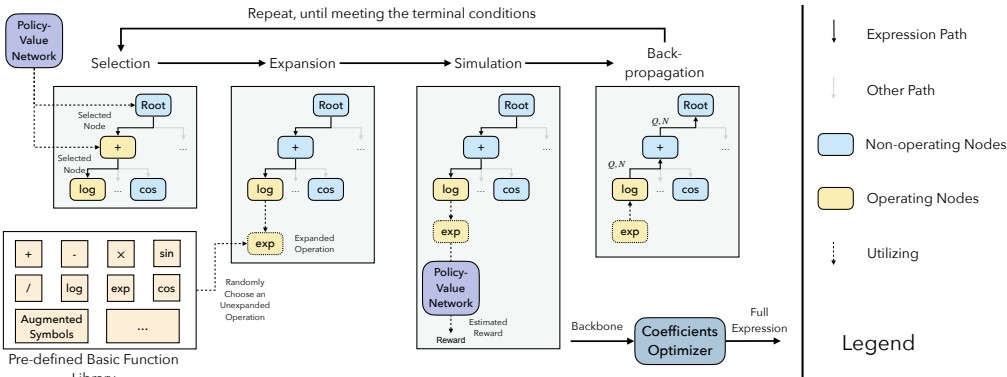

Figure 2: The overview of NEMoTS.

- **Function Library**: Crucial in MCTS's expansion phase, this library provides mathematical operations for building new nodes in the expression tree, such as addition, subtraction, and trigonometric functions.

- **Policy-Value Network**: Integral in MCTS's selection and simulation phases, this neural network evaluates the current expression and target time series. It selects promising nodes, assigns probabilities to operations, and scores the current state, enhancing decision-making in the search.

- **Coefficient Optimizer**: An independent part of NEMoTS, the optimizer zeroes in on finding optimal coefficients for the MCTS-created structure using efficient numerical methods. This finalizes the analytical expression, moving from a structural to a functional form.

### 3.2 MAIN PIPELINE

In this section, we will discuss how to generate the "backbone" of an expression using NEMoTS. It is important to note that each node in the tree maintains two variables: the total reward $Q$ and the visited count $N$.

### 3.2.1 SELECTION

Initially, a node designated as "Root" serves as a starting point for subsequent operations but does not contribute to expression generation.

The selection phase commences at the "Root" node and involves iterative selection of child nodes (potential mathematical operations from the pre-defined function library) until a partially expanded or unexpanded node is encountered. This process is a tree traversal employing a specific recursive strategy known as the Polynomial Upper Confidence Tree (PUCT). Formally, under tree state $S$, the PUCT score for a child node $a$ is given by:

$$Score(S, a) = Q(S, a) + c \cdot P(S, a) \cdot \frac{\sqrt{\sum_b N(S, b)}}{1 + N(S, a)}, \tag{2}$$

- $Q(S, a)$: The average or expected reward of choosing action $a$ in state $S$, based on its performance in similar situations.

- $P(S, a)$: The prior probability of picking action $a$ in state $S$, as estimated by the policy-value network. It gauges the potential worth of the action.

- $N(S, a)$: The number of times action $a$ has been chosen in state $S$, measuring how much the action has been explored.

- $c$: A constant that balances exploration (trying less-visited operations) and exploitation (using known high-rewarded operations). Higher $c$ favors exploration; the lower favors exploitation.

- $\sum_b N(S, b)$ The total number of visits to all actions $b$ in state $S$, used for normalizing exploration rewards.

Utilizing Eqn. 2, the child node with the highest score is chosen for further exploration until an unexplored node is reached. The path formed based on the PUCT is then converted into an expression.

### 3.2.2 EXPANSION

The expansion phase begins after the selection phase, focusing on a node that is either partially expanded or unexpanded, referred to as the "target node." This node is critical for adding new elements to the expression tree, central to the entire expansion process.

In this phase, a function from the library is randomly selected for expansion, following a uniform distribution. The selected operation, denoted as $a$ , is then integrated into the tree as a new node. This node's visit count $N(S, a)$ and total reward $Q(S, a)$ are initially set to zero. This method ensures equal opportunity for each function to be chosen, promoting fairness and variety in the exploration.

After integrating the new node, the process moves to the simulation phase. This stage is vital for the overall search strategy, involving simulations to foresee possible actions and outcomes. These simulations are key to shaping future decisions. The effectiveness of the simulation phase greatly affects the selection of future nodes and the development of the expression tree.

### 3.2.3 SIMULATION

In MCTS, particularly within our NEMoTS framework, the simulation phase is key for assessing the potential rewards of newly expanded nodes. This phase typically follows the expansion phase and starts from the most recently added node in the expression tree. It involves a rapid simulation method, often random, and continues **until the expression path surpasses a pre-defined length, the terminal condition**. The focus here is on quick evaluation rather than deep exploration.

NEMoTS diverges from traditional random simulations, which are time-intensive. Instead, we utilize the policy-value network's reward estimator for immediate reward estimations. This approach effectively evaluates the potential rewards of the current state and enhances the efficiency of the simulation process.

Crucially, during training, numerous random simulations are essential to provide supervised signals to the reward estimator. This ensures the policy-value network's scores are accurate and reflect real-world outcomes. This accuracy is vital for the effectiveness and precision of NEMoTS's simulation phase. The simulation concludes when the expression path reaches a predetermined length. Following the simulation phase, the generated expression path is transformed into an expression and further refined by the coefficient optimizer. The optimized expression is then assessed with the input signal as per Eqn. 1, leading to the back-propagation phase.

### 3.2.4 BACK-PROPAGATION

The back-propagation is a critical component in the MCTS, particularly in updating the decision-making mechanism. It follows the simulation phase and initiates at the node where simulation began, often the newly expanded node, and proceeds back to the root node, referred to as the "Root."

In this phase, for each node along the path from the start node of the simulation to the root, we update both the visit count $N(S, a)$ and the total reward $Q(S, a)$. These updates are influenced by the simulation outcomes and serve to adjust $Q(S, a)$, reflecting the new average or expected reward for an action $a$ in state $S$. Concurrently, $N(S, a)$ is incremented, indicating an additional visit to that child node. The reward data obtained at the end of the simulation is vital, as it helps in evaluating the long-term strategic benefits of the node.

Back-propagation is integral to refining the overall decision-making process. It enables the algorithm to better understand and adapt to the decision space through continuous learning. This phase ensures more efficient navigation of the expression tree and improves decision-making by reinforcing successful paths and reassessing less effective ones.

The process iterates through these four phases until the expression path reaches a pre-determined threshold. At this point, a preliminary "backbone" expression is formed, which still lacks specific

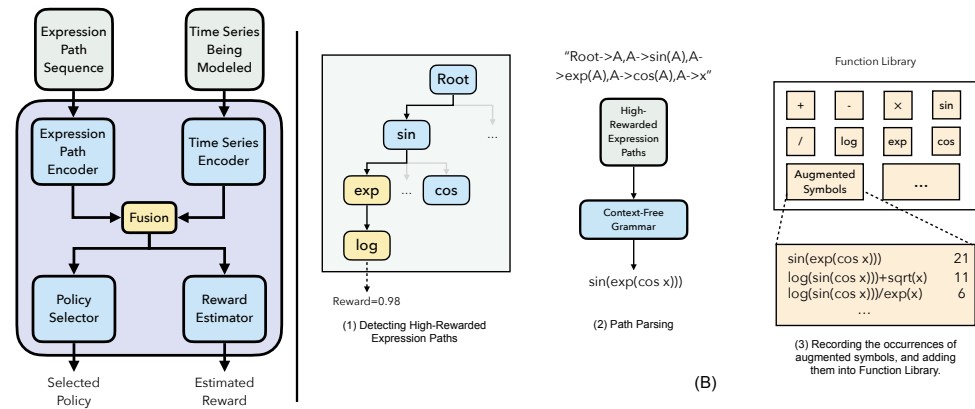

Figure 3: (A) The structure of policy-value network; and (B) The illustration of the symbolic augmentation strategy.

numerical coefficients. We then apply the Powell optimization method to this expression. This gradient-free algorithm is particularly suited for complex or non-differentiable functions, efficiently finding the function's minimum by updating search directions and conducting one-dimensional searches. This makes it an effective approach for problems where traditional gradient methods are not applicable, as discussed in Powell (1964).

### 3.3 MODEL TRAINING

The NEMoTS training process involves two primary components: refining the policy-value network and augmenting the function library through symbolic augmentation. The objective of optimizing the policy-value network is to improve its accuracy and efficiency in aiding the MCTS process. On the other hand, expanding the function library focuses on enriching the initial set of basic functions with more complex ones. This expansion serves a dual purpose: it provides encapsulation for expressions and customizes the library to more effectively match the specific dataset.

#### 3.3.1 POLICY-VALUE NETWORK

In NEMoTS, the policy-value network serves as a black-box model, handling two types of input: the expression path sequence and the input signals, namely the time series being modeled. Its outputs are twofold: the chosen operation and the estimated reward value, both based on the expression path sequence and input signals.

For our implementation, we employ Long Short-Term Memory (LSTM) networks Hochreiter & Schmidhuber (1997) to encode the expression path sequence. To process the input signal sequence, we use Temporal Convolutional Networks (TCN) Oord et al. (2016). We then concatenate the encoded representations from these two sources and use a Multilayer Perceptron (MLP) for further processing. This results in outputs for the three branches, as depicted in Fig. 3 (A).

A key aspect of our approach is optimizing the neural network. This optimization primarily focuses on designing an effective loss function that aligns with the MCTS process requirements.

- **Policy Selector**. The primary goal of the policy selector in NEMoTS is to generate accurate prior probabilities $P(S, a)$, crucial for forming a distinct probability distribution during the $Score(S, a)$ calculation, as indicated in Eqn. 2. This clarity in distribution is essential, guiding the model to selectively prioritize nodes for expansion more confidently. Accordingly, the optimization objective of this component is to minimize the Kullback-Leibler (KL) divergence between the prior probability distribution $P(S)$ and the posterior distribution $Score(S)$. Minimizing this divergence ensures that the model's predicted probability distribution closely mirrors the actual distribution of rewards, thereby significantly enhancing decision-making accuracy within the MCTS process.

$$Loss_{(PS)} = \sum_{a \in A} P(S, a) \log \left( \frac{P(S, a)}{Score(S, a)} \right), \tag{3}$$

where $A$ represents all valid nodes in the selection iteration.

- **Reward Estimator**. The reward estimator's function is to circumvent the intricate simulation phase, directly assessing the current state to produce a reward value. This makes it essentially a regression model. To train this component of the neural network effectively, we focus on minimizing the Mean Squared Error (MSE) between the output $\mathcal{R}'$ of the reward estimator and the simulated reward value $\mathcal{R}$. This minimization ensures that the reward estimator's predictions are as close as possible to the actual reward outcomes, thereby refining the model's efficiency in estimating rewards without the need for complex simulations:

$$Loss_{(RE)} = (\mathcal{R}' - \mathcal{R})^2 \tag{4}$$

Thus, optimizing the policy-value network equates to minimizing the loss of the above two parts:

$$Loss = \theta_1 Loss_{(PS)} + \theta_2 Loss_{(RE)}. \tag{5}$$

$\theta_1, \theta_2$ are coefficients used to balance these terms.

### 3.3.2 SYMBOLIC AUGMENTATION STRATEGY

In our study, we observed that relying solely on elementary functions in the function library often proved inadequate for accurately representing complex nonlinear dynamical systems in symbolic regression tasks. Consequently, the incorporation of more sophisticated composite functions became essential for precise time series representation. Contrary to previous approaches that depended on random selection in the MCTS simulation phase to identify high-reward expressions from a plethora of randomly generated composite functions Sun et al. (2022), our method takes a different path. We substituted the traditional random simulation with neural networks, which initially lacked the specialized ability to create specific composite functions for certain samples.

During the training phase, we implemented a strategy similar to frequent pattern mining Agrawal et al. (1993). This involved tracking expression paths and their corresponding high-reward composite functions. Typically, these expressions with high rewards emerged from random combination selections. According to the law of large numbers Durrett (2019), among these paths, some consistently received high rewards. After training, we analyzed the frequency of these high-reward paths. The most recurrent paths were deemed as optimal composite functions for our dataset and subsequently incorporated into our function library. This addition significantly enhanced our model's ability to represent time series and boosted the accuracy of modeling complex systems.

For practical applications, a novel element, termed 'augmented symbols', is introduced into the function library. During the expansion phase in MCTS, if the model selects this 'augmented symbol', it triggers a secondary sampling process. This process is based on a probability distribution reflecting the occurrence frequency of the top $k$ high-reward composite functions. Under this mechanism, a higher frequency of a specific 'augmented symbol' in the dataset indicates its more prevalent usage during training, suggesting a better fit for the current dataset, as illustrated in Fig. 3 (B). This approach markedly improves the model's adaptability to specific dataset characteristics, thus enhancing both the accuracy and efficiency of the symbolic regression process.

## 4 EXPERIMENTS

We conducted experiments on six real-world datasets to answer the following two core questions. For other questions that need to be addressed through experiments, please see the Appendix.

- Q1: How is the fitting ability and efficiency of NEMoTS?
- Q2: Are the expressions generated by NEMoTS reliable?

To address these questions, we will conduct two sets of experiments: fitting ability performance, and extrapolation analysis. In addition, other questions and experiment results will be presented in the Appendix.

## 4.1 DATASETS AND PRE-PROCESSING

We selected following univariate datasets for analysis, all datasets can be accessed through https://forecastingdata.org/: **The Electricity Demand from Victoria** (ELC), **Saugeen River Flow** (RiverFlow), **US Births**, **Weighted Influenza-Like Illness Percentage** (WILI), **Australian Daily Currency Exchange Rates** (ACER), **Atmospheric Pressure** (AP). The latter three datasets include ILI, Exchange rate, and Weather, as provided by previous research Zhou et al. (2021); Wu et al. (2021). We extracted relevant columns from these datasets. For symbolic regression tasks, we modified the ACER and AP datasets to include only the first 1000 timestamps, creating sub-datasets due to their time-intensive nature. The entire WILI dataset was utilized for comprehensive analysis owing to its smaller size.

## 4.2 EVALUATION METRICS

We evaluate algorithm performance using the coefficient of determination ($R^2$) and the correlation coefficient (CORR), which counteract the influence of data size. Efficiency is assessed through the Average Time Cost per sample in seconds (ATC), reflecting the algorithms' fitting ability and computational speed, useful for algorithm selection and optimization.

Given the significant random search in symbolic regression algorithms, we use actual time cost for efficiency evaluation. Though CPU tests were isolated to minimize external process interference, some variability in time cost due to computing environment fluctuations is expected. Despite their limitations, these measurements offer a general insight into the models' efficiency. For both $R^2$ and CORR, values nearing 1 indicate smaller regression error and a trend closer to actual values, respectively.

## 4.3 FITTING ABILITY PERFORMANCE

To address Q1, we focused on assessing the overall fitting ability. We used a sliding sampling method to divide time series data into samples of 36 and 72 time steps, for shorter and longer series respectively. In NEMoTS neural network training, 10% of each dataset was used for learning parameters, with the remaining 90% for testing. Importantly, NEMoTS's network is trained on the states processed by MCTS rather than directly on time series patterns, which obviates the need for a validation set. Table 1 provides a comparative performance summary. NEMoTS, once fully trained, excelled in 16 out of 18 metrics across six experimental sets involving three datasets and ranked second in the other two, highlighting its overall superiority. Our subsequent analysis will concentrate on two main aspects: performance, evaluated using the coefficient of determination $R^2$ and correlation coefficient CORR, and efficiency, assessed by average time cost per sample.

### 4.3.1 BASELINES

We will compare with the following methods: **Genetic Programming** (GP) Koza et al. (1994); gpl, **Multiple Regression Genetic Programming** (MRGP) Arnaldo et al. (2014), **Bayesian Symbolic Regression** (BSR) Jin et al. (2019), **Physics Symbolic Optimization** (PhySO) Tenachi et al. (2023), **Symbolic Physics Learner** (SPL) Sun et al. (2022).

For the coefficient-less backbones from these methods, we fit using least squares. Given the randomness in symbolic regression algorithms, which affects performance on certain samples, we exclude obviously abnormal metrics for fair comparison.

### 4.3.2 PERFORMANCE

The coefficient of determination $R^2$, a ratio of model error to average error, is a dimensionless metric assessing model fitting ability. Quantitatively, SPL and NEMoTS significantly surpass other models in the $R^2$ metric. SPL shows an average 203.04% improvement over models like GP, MRGP, BSR, and PhySO in $R^2$, while NEMoTS averages 229.21% improvement over all but SPL. This highlights SPL and NEMoTS's superior model fitting capabilities. The correlation coefficient (CORR) assesses the consistency between fitted and actual values. In this metric, SPL and NEMoTS also demonstrate considerable advantages. SPL achieves a 96.22% average improvement in CORR over the aforementioned models, and NEMoTS shows a 103.65% improvement over all but SPL. These

Table 1: Fitting Results. Both $R^2$ and CORR are dimensionless metrics, and ATC (Average Time Cost) by seconds.

| Dataset | Metrics | GP | | MRGP | | BSR | | PhySO | | SPL | | NEMoTS | |
|---|---|---|---|---|---|---|---|---|---|---|---|---|---|
| Seq. Length | | 36 | 72 | 36 | 72 | 36 | 72 | 36 | 72 | 36 | 72 | 36 | 72 |
| ELC | $R^2$ | 0.693 | 0.231 | 0.688 | 0.421 | 0.772 | 0.602 | 0.498 | 0.630 | 0.847 | 0.775 | **0.923** | **0.884** |
| | CORR | 0.834 | 0.313 | 0.792 | 0.449 | 0.810 | 0.683 | 0.562 | 0.663 | 0.903 | 0.861 | **0.951** | **0.909** |
| | ATC | 82.23 | 89.54 | 162.31 | 168.35 | 100.23 | 88.43 | 145.54 | 135.52 | 181.91 | 200.43 | **41.33** | **50.21** |
| RiverFlow | $R^2$ | 0.408 | 0.164 | 0.481 | 0.342 | 0.621 | 0.688 | 0.513 | 0.542 | 0.715 | 0.653 | **0.744** | **0.725** |
| | CORR | 0.591 | 0.193 | 0.552 | 0.503 | 0.717 | 0.702 | 0.552 | 0.559 | 0.782 | 0.710 | **0.774** | **0.751** |
| | ATC | 76.30 | 78.66 | 166.80 | 158.81 | 103.93 | 72.53 | 121.56 | 118.80 | 202.53 | 192.38 | **43.26** | **51.53** |
| USBirth | $R^2$ | 0.283 | 0.172 | 0.310 | 0.248 | 0.649 | 0.412 | 0.718 | 0.643 | 0.910 | 0.804 | **0.976** | **0.886** |
| | CORR | 0.493 | 0.233 | 0.443 | 0.310 | 0.801 | 0.508 | 0.762 | 0.682 | 0.923 | 0.871 | **0.981** | **0.916** |
| | ATC | 78.23 | 76.12 | 159.51 | 169.31 | 108.41 | 78.91 | 114.06 | 102.51 | 188.71 | 193.52 | **49.53** | **51.21** |
| WILI | $R^2$ | 0.302 | 0.287 | 0.377 | 0.267 | 0.541 | 0.089 | 0.488 | 0.375 | **0.937** | 0.863 | 0.923 | **0.890** |
| | CORR | 0.593 | 0.502 | 0.621 | 0.513 | 0.603 | 0.287 | 0.640 | 0.447 | **0.951** | 0.912 | 0.940 | **0.930** |
| | ATC | 93.35 | 90.51 | 203.44 | 198.51 | 113.23 | 57.72 | 103.51 | 106.53 | 223.25 | 231.01 | **28.13** | **42.18** |
| ACER | $R^2$ | 0.133 | 0.081 | 0.318 | 0.497 | 0.327 | 0.238 | 0.616 | 0.663 | 0.752 | 0.609 | **0.842** | **0.738** |
| | CORR | 0.215 | 0.178 | 0.422 | 0.531 | 0.541 | 0.364 | 0.701 | 0.715 | 0.838 | 0.780 | **0.857** | **0.831** |
| | ATC | 68.49 | 66.27 | 133.52 | 161.52 | 110.29 | 66.70 | 99.63 | 97.85 | 269.76 | 296.41 | **30.34** | **44.23** |
| AP | $R^2$ | 0.769 | 0.171 | 0.780 | 0.397 | 0.657 | 0.358 | 0.231 | 0.173 | 0.825 | 0.858 | **0.931** | **0.916** |
| | CORR | 0.716 | 0.378 | 0.707 | 0.566 | 0.628 | 0.461 | 0.615 | 0.286 | 0.869 | 0.906 | **0.955** | **0.943** |
| | ATC | 123.83 | 143.65 | 192.45 | 203.45 | 133.51 | 75.13 | 123.51 | 125.47 | 202.92 | 217.43 | **31.41** | **39.54** |

results underscore the effectiveness of SPL and NEMoTS in aligning model predictions with actual data trends.

SPL and NEMoTS's fitting proficiency partly stems from incorporating the MCTS algorithm. MCTS's design calculates historical returns to efficiently select high-rewarded operations and expressions. It also abstracts expressions as tree structures, ensuring validity and reducing numerical problems. NEMoTS outperforms SPL, owing to its policy-value network that learns from extensive data, enabling more focused selection and expansion phases in search of quality expressions. However, MCTS's inherent randomness can impact model performance stability. Despite this, the MCTS-based design of NEMoTS and SPL offers robust fitting ability for symbolic regression.

### 4.3.3 EFFICIENCY

We assessed algorithm efficiency by evaluating the average time cost per sample. The results show NEMoTS with a substantial reduction in average time cost compared to other methods (GP, MRGP, BSR, PhySO, SPL), achieving about a 68.06% improvement. This efficiency gain primarily arises from incorporating the policy-value network. Unlike other models, NEMoTS bypasses numerous simulations and search steps, using its neural network for direct assessment of the current expression. This approach, by eliminating the need for time-intensive simulations and searches, markedly improves NEMoTS's efficiency. Consequently, NEMoTS exhibits a significant competitive edge in time efficiency, proving advantageous for large-scale or time-sensitive tasks.

### 4.4 EXTRAPOLATION

To answer Q2, we undertake an extrapolation task, commonly known as short-term prediction, which is a pivotal aspect of time series analysis. The fundamental idea is to evaluate whether an expression, derived through symbolic regression, can accurately predict the future evolution of a time series. Successfully doing so would demonstrate that the expression has adeptly captured the core pattern inherent in the time series. This critical evaluation allows us to substantiate the interpretative reliability of our model, offering insights into its ability to decipher and project data trends.

Practically, we will implement this extrapolation task on the same three datasets previously mentioned. In each case, we will analyze time series data covering 30 time steps, with the objective of forecasting the subsequent 6 time steps. This methodology is designed to rigorously test the model's proficiency in short-term forecasting. By doing so, we aim to comprehensively evaluate the model's capacity to not only understand but also accurately project the underlying patterns of time series data. This approach is particularly useful in determining the model's effectiveness in navigating

and interpreting complex data sequences, thereby providing a robust assessment of its predictive capabilities and reliability in real-world scenarios.

In this section, we will use the following time series prediction methods to compare: **Auto-Regressive Integrated Moving Average** (ARIMA) Moreira-Matias et al. (2013); **Support Vector Regression** (SVR) Awad et al. (2015); **Recurrent Neural Networks** (RNN) Wen et al. (2017); **Temporal Convolutional Networks** (TCN) Oord et al. (2016); **Neural Basis Expansion Analysis** (NBeats) Oreshkin et al. (2019).

It is important to note that we do not intend to introduce overly complex prediction models in this context. This is because the purpose of the extrapolation task is merely to verify whether the derived expression captures the intrinsic dynamics of the time series. The primary objective of this study is to offer insights into interpretability, rather than to focus on prediction.

Table 2: Extrapolation performance across different baselines for various datasets.

| Dataset | Metrics | ARIMA | SVR | RNN | TCN | NBeats | NEMoTS |
|---------|---------|-------|-----|-----|-----|--------|--------|
| ELC | $R^2$ | -0.931 | -0.785 | 0.468 | 0.354 | 0.503 | **0.703** |
|  | CORR | 0.434 | 0.223 | 0.503 | 0.447 | 0.462 | **0.715** |
| RiverFlow | $R^2$ | -0.925 | -0.903 | 0.313 | 0.235 | 0.496 | **0.579** |
|  | CORR | 0.166 | 0.118 | 0.170 | 0.319 | 0.523 | **0.617** |
| USBirth | $R^2$ | -1.193 | -1.700 | 0.182 | 0.225 | 0.461 | **0.528** |
|  | CORR | 0.345 | 0.223 | 0.317 | 0.474 | 0.488 | **0.529** |
| WILI | $R^2$ | -1.512 | -1.085 | 0.231 | 0.214 | 0.453 | **0.617** |
|  | CORR | 0.234 | 0.181 | 0.313 | 0.307 | 0.312 | **0.566** |
| ACER | $R^2$ | -1.187 | -0.943 | 0.435 | 0.115 | 0.393 | **0.479** |
|  | CORR | 0.126 | 0.228 | 0.133 | 0.203 | 0.465 | **0.617** |
| AP | $R^2$ | -1.433 | -1.854 | 0.215 | 0.336 | 0.461 | **0.628** |
|  | CORR | 0.445 | 0.423 | 0.417 | 0.518 | **0.658** | 0.629 |

Overall, the NEMoTS model demonstrates superior performance in most scenarios, particularly excelling in the $R^2$ metric, where it achieves the highest scores across all three datasets. In contrast, the ARIMA and SVR models generally exhibit poor performance, especially in the $R^2$ metric, where these models show negative values in most datasets, indicating lower prediction accuracy. On the AP dataset, all models exhibit relatively high CORR values, suggesting that their predictions are more closely correlated with actual outcomes.

The performance of all models on two key metrics is only moderately satisfactory, mainly due to two factors: the challenging nature of the datasets, which are non-stationary and non-periodic, making prediction difficult; and the limited training data, with only 10% of samples used from 1000 timestamps, leading to insufficient training of neural network-based methods. These issues combine to limit the models' ability to accurately forecast future trends.

Despite these challenges, NEMoTS outperforms other prediction models, demonstrating not only its strength in prediction and extrapolation tasks but also its ability to capture key evolutionary traits of time series data. Its accuracy in identifying underlying dynamics, despite various influencing factors, highlights its effectiveness. This success in recognizing intrinsic patterns showcases the model's robustness in handling complex data and reinforces its value in practical applications where understanding and predicting data trends are essential.

## 5 CONCLUSION

In this study, we apply symbolic regression techniques to time series analysis, improving its interpretability by extracting analytical expressions. To tackle the large search space in symbolic regression, we adopt Monte-Carlo Tree Search (MCTS), which narrows down the search space and ensures expression validity through structured constraints. We enhance the efficiency and generalization by integrating neural networks, which guide MCTS and replace conventional simulations, boosting efficiency and learning capabilities. Additionally, our Symbolic Augmentation Strategy captures and utilizes common composite functions, enhancing the fitting ability. Our extensive tests on six real-world datasets demonstrate the superior performance, efficiency, reliability, and interpretability in time series analysis.

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

## A  RELATED WORKS

### A.1  SYMBOLIC REGRESSION

Symbolic Regression (SR) emerges as a sophisticated technique in regression analysis, uniquely characterized by its ability to simultaneously identify both the parameters and the functional forms of equations that best describe given datasets. This method stands apart from traditional regression approaches, such as linear or quadratic regression, by offering a more holistic and adaptable framework for data analysis Angelis et al. (2023); Tohme et al. (2022); McConaghy (2011).

Central to SR's methodology is its data-driven nature, which operates independently of preconceived models or theories about the system under investigation Kabliman et al. (2019). This characteristic is particularly advantageous when dealing with datasets that encompass ambiguous or complex relationships. SR's capacity to unearth these intricate associations not only provides innovative solutions to challenging problems but also fosters a deeper comprehension of systems that are only partially understood. Furthermore, SR's prowess in generating closed-form mathematical expressions renders it an invaluable asset in the realm of generalizable AI. Its compatibility with various modeling tools, such as finite element solvers, underscores its versatility and broad applicability Gilpin (2021).

One of the most notable achievements of SR is its ability to rediscover and validate fundamental physical laws, exemplified by its replication of Newton's law of gravitation through purely data-driven means Lemos et al. (2023); Liu & Tegmark (2021). This capacity underscores SR's potential in empirically grounding theoretical constructs. However, it's important to acknowledge the challenges inherent in this method. The risk of deriving spurious results due to oversimplified datasets or the absence of robust evaluation metrics is a notable concern Matsubara et al. (2022). SR shows particular efficacy in analyzing complex, nonlinear dynamic systems, distilling governing expressions directly from observational data. Despite these strengths, SR's effectiveness can be compromised by factors such as data scarcity, low fidelity, and noise. Nevertheless, the application of Bayesian methodologies has shown promise in mitigating these limitations Wilstrup & Kasak (2021). Compared to several machine learning models, SR has demonstrated superior performance, especially in scenarios involving small datasets. However, a significant drawback of SR is its computational intensity, as the evaluation of numerous potential equations can be time-consuming. This aspect makes SR more suitable for scenarios with a limited number of input parameters Cao & Zhang (2022).

### A.2 MONTE-CARLO TREE SEARCH

Monte-Carlo Tree Search (MCTS) is a prominent method in artificial intelligence for making optimal decisions by sampling randomly in the decision space and constructing a search tree based on the outcomes Browne et al. (2012). This approach has significantly impacted AI, particularly in fields that can be modeled as trees of sequential decisions, like games and planning problems Silver et al. (2016; 2017).

A typical MCTS consists of four main stages:

1. **Selection**: The process begins at the root node and involves selecting successive child nodes until a leaf node is reached. This selection is based on a tree policy, typically the Upper Confidence Bound (UCB) applied to trees, which balances exploration and exploitation. The UCB formula takes into account both the average reward of the node and the number of times it has been visited;

2. **Expansion**: Once a leaf node is reached, one or more child nodes are added to expand the tree, depending on the available actions. This step is crucial for exploring new parts of the search space;

3. **Simulation**: This involves simulating a play from the newly added nodes to the end of the game using a default or random policy. The simulation phase is where MCTS diverges from traditional tree search methods, as it involves playing out random scenarios to get an estimate of the potential outcome from the current state.

4. **Back-propagation**: In the final phase, the results of the simulation are propagated back up the tree. The nodes visited during the selection phase are updated with the new information, typically involving updating their average reward based on the outcome of the simulation and incrementing the visit count.

MCTS is fundamentally a decision-making algorithm grounded in search and probability rather than a conventional machine learning algorithm. However, due to its complex heuristic rules, the algorithm encounters several challenges: (1) Its intricate simulation process requires numerous simulations, resulting in high complexity and low efficiency; (2) As a decision-making algorithm, MCTS operates only at the instance level and lacks the capacity to learn from large-scale data, thus missing the opportunity for inductive learning from extensive datasets to enhance performance. This paper addresses these two critical issues. We adapt MCTS to the specific needs of symbolic regression tasks, focusing on discovering expressions while also aiming to enhance MCTS's efficiency in this domain. Additionally, we enable MCTS to acquire learning capabilities, allowing it to optimize itself with vast amounts of data.

## B COMPLEXITY DISCUSSION: NEMoTS AND NAIVE MCTS

### B.1 OVERVIEW

This section compares the complexity of NEMoTS (Neural-Enhanced Monte Carlo Tree Search) and naive MCTS in symbolic regression tasks. We consider the case where the maximum expression length is $L$, and the size of the symbol set is $|A|$. Since both NEMoTS and naive MCTS use probabilistic search methods, it is challenging to precisely determine their complexity for specific problems. Therefore, we analyze their complexity based on the worst-case scenario.

For complexity testing of these methods, see Section 4.3.

### B.2 COMPLEXITY ANALYSIS OF NAIVE MCTS

MCTS is a probabilistic search algorithm where each node in the tree represents a symbolic operation. For a maximum expression length of $L$ and a symbol set of size $|A|$, in the worst case, the search tree is a $|A|$-ary tree with a maximum number of nodes:

$$N = \sum_{i=0}^{L} |A|^i = \frac{|A|^{L+1} - 1}{|A| - 1}$$

Therefore, we conclude:

- **Space Complexity**: In the worst case, the search tree is fully expanded, containing approximately $|A|^L$ nodes. Thus, the space complexity is $\mathcal{O}(|A|^L)$. Note that such full expansion may not be common in practical applications due to pruning and heuristic strategies that limit tree growth.

- **Time Complexity**: Analyzing the main phases of MCTS:
  - **Selection Phase**: Traverses from the root to a leaf node, with a maximum path length of $L$, so the time complexity is $\mathcal{O}(L)$.
  - **Expansion Phase**: Adds a new node, with time complexity $\mathcal{O}(1)$.
  - **Simulation Phase**: Performs random simulation steps, with the number of steps $S$. The time complexity is $\mathcal{O}(S)$, where $S$ is usually much larger than $L$.
  - **Backpropagation Phase**: Updates statistical information along the path from the current node to the root, with time complexity $\mathcal{O}(L)$.

  Combining the above phases, the time complexity per iteration is:

$$T_{\text{MCTS}} = \mathcal{O}(L) + \mathcal{O}(1) + \mathcal{O}(S) + \mathcal{O}(L) = \mathcal{O}(2L + S)$$

### B.3 COMPLEXITY ANALYSIS OF NEMoTS

NEMoTS introduces neural networks into MCTS to improve the efficiency of the simulation phase. Its complexity analysis is as follows:

- **Space Complexity**: The search tree structure of NEMoTS is the same as MCTS, so the space complexity remains $\mathcal{O}(|A|^L)$.

- **Time Complexity**: Analyzing the main phases of NEMoTS:
  - **Selection Phase**: Time complexity is $\mathcal{O}(L)$.
  - **Expansion Phase**: Time complexity is $\mathcal{O}(1)$.
  - **Simulation Phase**: Uses a pre-trained neural network for evaluation, assuming the inference time of the neural network is $T_{\text{NN}}$. Although the inference time is not strictly constant, it can be approximated as $\mathcal{O}(1)$ if the network is of moderate size.
  - **Backpropagation Phase**: Time complexity is $\mathcal{O}(L)$.

  Therefore, the time complexity per iteration of NEMoTS is:

$$T_{\text{NEMoTS}} = \mathcal{O}(L) + \mathcal{O}(1) + \mathcal{O}(T_{\text{NN}}) + \mathcal{O}(L)$$

If $T_{\text{NN}}$ can be approximated as constant, then:

$$T_{\text{NEMoTS}} = \mathcal{O}(2L)$$

Compared to MCTS, NEMoTS reduces the time complexity of the simulation phase from $\mathcal{O}(S)$ to $\mathcal{O}(T_{\text{NN}})$, significantly improving efficiency.

### B.4 QUANTITATIVE ANALYSIS OF EFFICIENCY IMPROVEMENT

The simulation phase is usually the most time-consuming part of MCTS. Suppose in naive MCTS, the number of simulation steps is $S$, e.g., $S = 200$. In NEMoTS, the simulation phase uses a neural network evaluation, assuming inference time $T_{\text{NN}} \approx 1$.

The efficiency improvement ratio can be expressed as:

$$\text{Efficiency Improvement Ratio} = \frac{T_{\text{MCTS}} - T_{\text{NEMoTS}}}{T_{\text{MCTS}}} = \frac{(2L + S) - (2L + T_{\text{NN}})}{2L + S} = \frac{S - T_{\text{NN}}}{2L + S}$$

When $L = 20$, $S = 200$, and $T_{\text{NN}} = 1$:

$$\text{Efficiency Improvement Ratio} = \frac{200 - 1}{40 + 200} = \frac{199}{240} \approx 82.9\%$$

This indicates that under these parameters, NEMoTS can reduce the time complexity per iteration by approximately 82.9%, significantly enhancing the algorithm's efficiency.

### B.5 ASSUMPTIONS AND CONSIDERATIONS

To make the above derivation more rigorous, we need to clarify the following assumptions:

- **Inference Time of Neural Network**: Assuming that the inference time $T_{\text{NN}}$ of the neural network can be approximated as constant. This is reasonable when the neural network is of moderate size and inference is fast, but for large networks, this may need to be reevaluated.
- **Worst-Case Analysis**: Our complexity analysis is based on the worst-case scenario where the search tree is fully expanded. In practical applications, pruning and heuristic strategies may significantly reduce the actual space and time complexity.
- **Importance of the Simulation Phase**: In MCTS, the simulation phase is used to evaluate the potential value of nodes. NEMoTS replaces random simulations with a neural network, and the prediction accuracy of the neural network is crucial to the algorithm's performance. If the neural network's predictions are inaccurate, it may affect the effectiveness of the search.
- **Specificity of Symbolic Regression Tasks**: In symbolic regression tasks, the search space is enormous, but not all expressions are valid. Using heuristic methods like neural networks can effectively guide the search and improve efficiency.

### B.6 CONCLUSION

By introducing neural networks, NEMoTS significantly reduces the time complexity while maintaining the same space complexity as naive MCTS, particularly reducing the overhead of the simulation phase. Under the parameters $L = 20$ and $S = 200$, NEMoTS can reduce the time complexity per iteration by approximately 82.9%, demonstrating its advantages in symbolic regression tasks.

## C ALGORITHM PROCESS

The detailed algorithm process is shown as follows.

**Algorithm 1:** Process of Neural-Enhanced Monte-Carlo Tree Search (NEMoTS)

**Input:** Time series data $\mathcal{D}$, function library, policy-value network
**Output:** Full expression

**Procedure:** NEMoTS
 Initialize root node as "Root"
 **while** (not the terminal condition) **do**
  $leaf \leftarrow$ Select($root$)
  $child \leftarrow$ Expand($leaf$)
  $reward \leftarrow$ Simulate($child$)
  Back-Propagate($child, reward$)
  Symbolic Augmentation Strategy($child, reward$)
 **end while**
 $backbone \leftarrow$ Generate backbones from $root$
 $fullExpression \leftarrow$ Optimize Coefficients($backbone$)
**end Procedure**

**Function** Select($node$):
 **while** ($node$ not fully expanded) **do**
  $node \leftarrow$ PUCT($node$)

**end while**
    **return** $node$

**Function** Expand($node$):
    % Get expansion probabilities using policy-value network
    Randomly select an operation from function library
    **if** (select operation is the augmented symbols) **then**
        Perform secondary sampling from augmented symbols
    **end if**
    Add new child node to $node$ with selected operation
    **return** new child node

**Function** Simulate($node$):
    Evaluate $node$ using reward estimator
    **return** evaluation reward

**Function** Back-Propagate($node, reward$):
    **while** ($node$ not null) **do**
        Update $node$'s total reward $Q$ and visit count $N$
        $node \leftarrow$ parent of $node$
    **end while**

**Function** PUCT($node$):
    Calculate PUCT score for each child of $node$
    **return** child with highest PUCT score

**Function** Optimize Coefficients($expression$):
    Optimize $expression$ using gradient-free methods (e.g., Powell's method)
    **return** optimized $expression$

**Function** Symbolic Augmentation Strategy($node, reward$):
    Record expression path and $reward$ of $node$
    Update symbol enhancement records based on $reward$
    Adjust function library based on records

## D  CASE STUDIES

In this part, we will answer the question: **How can the interpretability of NEMoTS be analyzed?** To this end, We compare NEMoTS-derived symbolic regression expressions with actual dataset data, as shown in Fig. D. These visualizations demonstrate NEMoTS's effectiveness in fitting complex real data with succinct mathematical models, capturing time series trends. This confirms NEMoTS's capability in fitting intricate data and its efficiency in trend extraction and representation, providing insightful analysis of time series. Additionally, our analysis includes predictive assessments on future data not used in the fitting, shown in sections with a gray background in the figures. NEMoTS not only accurately fits existing data but also forecasts future trends reliably. This indicates NEMoTS's strong predictive power, enhancing the value of its identified expressions. The ability of NEMoTS to accurately predict future trends signifies its interpretability and reliability. This accuracy is crucial for deeper understanding and forecasting of time series data, especially in domains that demand precise data prediction and interpretation.

## E  ABLATION STUDIES

In this part, we will answer the question: **What enhancements in performance and efficiency do NEMoTS achieve through improvements?** To this end, we carried out ablation studies by individually omitting the policy-value network and the symbolic augmentation strategy from NEMoTS to evaluate their impacts on efficiency and performance. Specifically, we tested the model under four

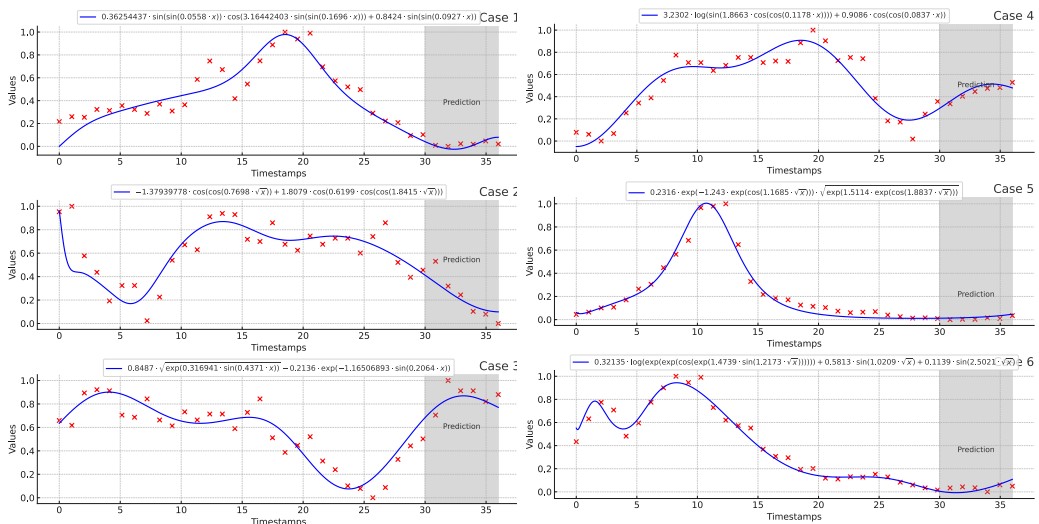

Figure 4: Case studies

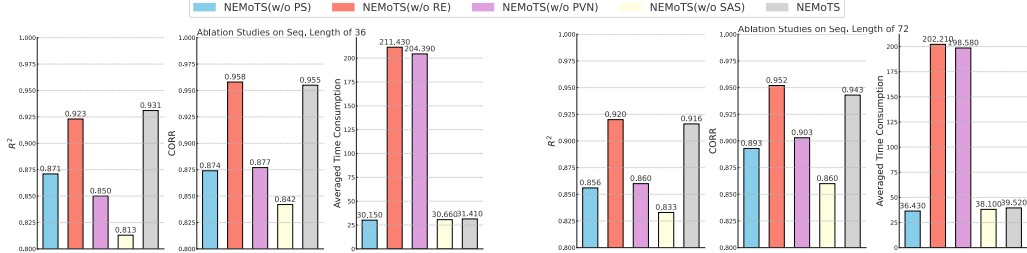

Figure 5: The results of ablation studies.

scenarios: 1) NEMoTS without the policy selector (NEMoTS(w/o PS)), utilizing the UCB score for node selection; 2) NEMoTS without the reward estimator (NEMoTS(w/o RE)), employing a random strategy for reward simulation; 3) NEMoTS lacking the entire policy-value network (NEMoTS(w/o PVN)); and 4) NEMoTS minus the symbolic augmentation strategy (NEMoTS(w/o SAS)). The experiments were conducted on the AP dataset, with sequence lengths of 36 and 72. Results of the ablation study are displayed in Fig. 5.

### E.1 POLICY SELECTOR

The policy selector in NEMoTS plays a critical role in evaluating the expression sequence and time series data from the root node, essential for guiding the selection of child nodes in Monte-Carlo Tree Search (MCTS). It effectively narrows the search space, enhancing efficiency and precision. Ablation study results emphasize its significance. Its removal results in a 6.45% decrease in the coefficient of determination ($R^2$) and a 6.89% reduction in the correlation coefficient (CORR). This suggests that without the policy selector, MCTS is less efficient at identifying optimal operations, negatively affecting overall performance. Notably, the absence of the policy selector also leads to a lower average time cost per sample. In the ablation model without the policy selector, the standard prior distribution is replaced by a uniform distribution, thereby reducing the need for neural network operations and shortening processing time.

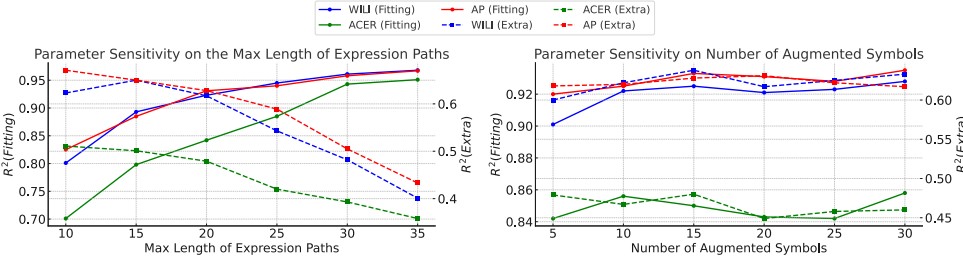

Figure 6: Parameter sensitivity. Different colors represent different datasets. The solid lines illustrate the fitting ability, referring to the left vertical axis, while the dashed lines indicate the extrapolation ability, referring to the right one.

### E.2 REWARD ESTIMATOR

The reward estimator is pivotal for evaluating the current state's reward score, thereby bypassing the need for extensive simulations typical in traditional MCTS. Our ablation study shows that while its removal slightly improves model performance, it substantially increases the time cost. This higher time cost is due to traditional MCTS's dependence on numerous, time-intensive simulations. The slight improvement in performance may be because neural network-based reward estimations aren't as precise as those from simulations, which could impact the model's precision. Essentially, the neural network offers a faster but potentially less accurate scoring method compared to traditional simulations. Therefore, removing the reward estimator leads to a minor improvement in performance but a significant increase in time cost. This trade-off underlines the importance of balancing performance gains and time efficiency in practical scenarios, especially when deciding on the use of a reward estimator.

### E.3 POLICY-VALUE NETWORK

In the Monte-Carlo Tree Search (MCTS) process, each element of the policy-value network – the policy selector and the reward estimator – is vital, greatly influencing decision accuracy, expansion efficiency, and precision in reward assessment. Without the policy selector, there's a decline in efficiency for selecting optimal operations. Conversely, removing the reward estimator could enhance performance but at the expense of increased time cost. The removal of the entire policy-value network underscores the combined impact of these components. This exclusion can result in varied model performance, highlighting the need for a balanced interplay among the different elements in the model.

### E.4 SYMBOLIC AUGMENTATION STRATEGY

Symbolic augmentation strategy enhances the function library by incorporating high-reward composite functions developed during training. These functions capture specific dataset patterns more comprehensively than basic elementary functions, offering a fuller understanding of time series characteristics.In standard Monte-Carlo Tree Search (MCTS), identifying complex patterns is often difficult. However, the symbolic augmentation strategy aids in recognizing and using these complex patterns more effectively, significantly boosting model performance. Without this strategy, there's a potential for underutilization of these intricate patterns, which could lead to a marked decrease in performance. In summary, the symbolic augmentation strategy is key in NEMoTS, as it includes high-reward composite functions in the function library. This enhances the model's ability to identify and articulate complex time series patterns, thus improving accuracy and efficiency in both analysis and prediction.

## F PARAMETER SENSITIVITY

In this part, we will answer the question: **How will different hyperparameters in the model affect its performance?** To this end, we focus on two hyperparameters closely tied to the model: the

maximum length of expression paths and the number of augmented symbols ($k$) in the Symbolic Augmentation Strategy. The experiment is conducted on three datasets: WILI, ACER, and AP, with detailed comparisons shown in Fig. 6, including fitting and extrapolation performance. The results indicate that as the max length of expression paths increases, fitting ability improves significantly, which is expected since longer paths allow for more complex analytical expressions, enabling a more accurate fit to the data. However, this improvement shows diminishing returns beyond a certain threshold, approximately 20 in our experiment, where further increases yield limited performance gains. Conversely, as the max length increases, extrapolation ability declines, more noticeably as the length extends. This decline is due to the overfitting phenomenon, where the analytical expression fits the training data too well, reducing its generalization and extrapolation capabilities.

Regarding the number of augmented symbols ($k$), this hyperparameter does not have a direct impact on the model, affecting neither fitting nor extrapolation ability. We found that augmented symbols, derived from frequent pattern mining, are ordered by occurrence frequency, following a power-law distribution. As the augmented symbols expands, the newly added symbols appear less frequently, resulting in the modeling primarily relying on the highest frequency symbols. This explains why expanding the augmented symbols library does not significantly influence performance.

