# OpenReview forum: "A GENERALIZABLE AND EFFICIENT SYMBOLIC RE- GRESSION METHOD FOR TIME SERIES ANALYSIS"
_ICLR.cc/2025/Conference — ICLR 2025 Conference Withdrawn Submission_

### Official Review · Reviewer_UU37 · 2024-10-30

**Soundness:** 2
**Presentation:** 1
**Contribution:** 2
**Rating:** 3
**Confidence:** 4

**Summary:**

This contribution is about a somewhat new method for time series analysis using symbolic regression, called Neural-Enhanced Monte-Carlo Tree Search (NEMoTS). The authors claim that this model has several benefits including enhanced understanding as it addresses limitations of traditional quantitative methods by offering explicit mathematical expressions for time series, providing intuitive and qualitative insights. Their approach frames time series analysis as a combinatorial optimization problem, using reinforcement learning and symbolic regression to derive expressions for non-linear dynamics. Furthermore there is integration of neural networks into the picture since NEMoTS claims to enhance Monte-Carlo Tree Search (MCTS) with neural networks, improving generalizability and computational efficiency by guiding the selection process and replacing complex simulations. Symbolic augmentation strategies seems to be Introduced, inspired by frequent pattern mining, to improve the simulation phase, enhancing the model's fitting capabilities.

The authors perform experiements that seemingly demonstrate superior performance, efficiency, and reliability across six real-world datasets, outperforming existing symbolic regression methods.

Overall, the paper combines several ideas into a "new package" and there is some credit there to be given as to experimenting with this. However, the paper could benefit from significantly clearer writing. It contains many buzzwords and statements that would be stronger with more detailed explanations and supporting evidence. The introduction of concepts is somewhat vague, and there is a noticeable lack of mathematical content. As a matter of fact, the lack of mathematical content is a severly limiting factor since not only no theory is provided but also no hints of it are discussed clearly.

While in general, the paper is quite descriptive, it would be helpful to have more concrete details to better understand the experiments, as their reproduction is currently challenging. Simply put, not enough information is provided.

The model introduced is based on Monte Carlo Search Trees and neural networks. However, the explanation of its functionality is quite superficial. It would be beneficial to have a clearer understanding of how the reward estimation neural network contributes to improvements. Additionally, a comparison with a standard Monte Carlo Search Tree for symbolic regression as a baseline would provide valuable insights.

**Strengths:**

The combination of the ideas is not necessarily bad. It could be the case that, at high-level, NEMoTS can be of some good value.

**Weaknesses:**

Let me dive into more descriptive feedback:

Introduction:
Besides from the bad grammar, the high amount of marketing words, and logical flaws in some places (remove the simulation aspect from MCST with advanced fitting to streamline MCST which has a simulation part?), the introduction brings the topic of what and why changes are observed in TS. Pretty much like trying to bring some causality into TS analysis, and trying to use an improved version of Monte Carlo search trees to fix it. There is no theoretical justification.

Problem definition:
Vague definition "Find an analytical expression and evaluate it", but what for?
The factor in the reward function is "slightly less than 1", but by how much? When is it too much, when is it not enough less than 1?

Neural-enhanced MCTS:
The writing is below par, every component of the new framework is introduced as either "key", "crucial" or "integral".
Model overview: Also elementes of superficial writing, difficult to really understand how each component interacts with each other.
Main pipeline:
- Selection: The authors introduce the PUCT score for a child node of a tree of state S, in order to start the exploration with the highest scoring node.
- Expansion: Adds a random function from the function library in the sequence, ok.
- Simulation: They say they use the policy value network to estimate the reward and claim it is better, but don't say how they do it nor provide any real justification on why.
- Backprop: Talks about how integral and critical this step is, but stays vague and provide no mathematical basis. Mentions Powell optimisation, but doesn't say how it is used or why another suitable optimisation isn't used instead.

Model training:
Policy-value network:
They use LSTM for the expression path sequence and Temporal Convolutional Networks for the input signal sequence. Concat them and use an MLP. They say the MLP outputs 3 values which are independent, but their illustration shows only 2 outputs? And is it even a good idea to predict 2 things at the same time with one MLP? They will end up interfering with each other during the learning process.
They introduce the loss they will use for the problem. It is a KL divergence coupled with an MSE between estimated reward and expected reward, ok.
Symbolic augmentation strategy:
Besides from saying that the author of a Cambridge book on probabilities created the Law of Large Numbers in 2019, this part is rather superficial. It says how much augmented expressions add value, without giving anything tangible. We still don't know what an augmented expression concretely is after reading this part (even after reading the whole paper tbh).

Experiments:
No word on the protocol. No word on the model. We do not know anything about hyperparameters, model size, etc.
No code shared, nothing reproducible.
They get SOTA performances by training neural networks on 10% of a dataset and testing it on the remaining 90%.
They mention R2 and CORR, without telling how they were calculated (what result from the model they exactly inject in those metrics to check the benchmark)
However they are the best overall, sometimes oddly better than other models on some metrics (all models have an average time cost above 100 seconds while the paper's model has one of 40 seconds)

For the reasons above I cannot be but suspicious of the correctness/soundness of these results. These results and ideas are very vague, irreproducible, at least in the current form, and yet so good compared to other sota models.

Conclusion:
"In conclusion, our model is better"
The authors don't say anything else.

So, to sum it up, I am highly skeptical on the way all these ideas where captured. While domain expertise to some extent has been put forward it sounds like a mixture of ideas that, say, some sophisticated pretrained algorithm would provide, with some good ability of putting all if together. Much works needs be done in order for this paper to be on par of the ICLR venue.

**Questions:**

See above. All questions are mentioned within the "Weaknesses" section.

**Details Of Ethics Concerns:**

There are certain sections of the contribution that looks like maybe not human generated. Phrases and words such as
1."Unlike traditional quantitative methods..."
2.".. delves deeper.."
3,"... learn from larger datasets..."
to mention a few, look a bit too common.

---

### Official Review · Reviewer_8Yh8 · 2024-11-03

**Soundness:** 1
**Presentation:** 2
**Contribution:** 2
**Rating:** 3
**Confidence:** 3

**Summary:**

Disclaimer: I am not very familiar with symbolic regression, especially in the time series domain.

Summary: This paper explores using symbolic regression for time series analysis. To fit a mathematical expression to a time series, a library of basic operations are iteratively used to construct a tree using MCTS with neural network based policy and reward estimation. They also add high reward paths into the library to make the search more efficient. The experiments on 6 datasets show this symbolic regression method achieves both better fitting quality and forecasting (extrapolation) performance.

**Strengths:**

- I think it is an interesting topic and I am curious about how such an approach performs in both fitting and extrapolation scenarios.
- It’s a novel idea to search for mathematical expressions for time series analysis using MCTS, at least to me.

**Weaknesses:**

- Extensive details are missing, thus, hurting the soundness of the paper. See more detail in the questions.
- There is no discussion of the limitation of the method. For example, how do the initial library functions are derived? How large the context window can be?

**Questions:**

- $\eta$ defines the amount of penalization with larger expressions, how to set this hyperparameter? What’s the value being used? Why choose that value? Whats’ the choice of $C$ in eq(2) in the experiments?

- How do the authors convert the tree into a sequence and back? Are you using Prufer sequence or something else?

- What is a tree state? What does it contain and what does not?

- In eq(3), $score(S, a)$ uses $P(S, a)$, how to compute the loss exactly? How to compute $score(S, a)$ as groundtruth?

- What is the training process of policy-value networks and the reward estimators? Are the training dataset dependent or independent? As in Figure (3a), it needs expression and time series. Where does the training expression come from? What is the architecture of the LSTM?

- What is the reason the authors skip long time series in their evaluation? Given an expression and generating observations should be very fast. Is it because the LSTM is not efficient for long sequences? Nevertheless, not being able to process a relatively long context window (a few hundreds) is a weakness of the method.

- Is the time for training the neural network included in the time reported in Table 1?

- How well is the powell method doing in these experiments?

- About augmentation, there are multiple paths in a tree and how to choose the paths or a segment of the path to be added to the library? Based on what threshold? Even in the Appendix C, only the ideas are given, not the detail.

- ARIMA’s performance is really bad on R2, is it because too short context is given to the model? Does the author use Auto-ARIMA or just plain ARIMA with some default hyperparameter?

- Following the above, how are the baseline methods used? Are they being used correctly and fairly? The results on Table 2 suggest a very big jump for the proposed methods on some datasets. Any thoughts on why such big improvements can be made?

---

### Official Review · Reviewer_2Q6M · 2024-11-04

**Soundness:** 3
**Presentation:** 1
**Contribution:** 3
**Rating:** 5
**Confidence:** 2

**Summary:**

The authors in this paper are interested in symbolic regression for time series data. In this aim, the authors introduce Neural-Enhanced Monte-Carlo Tree Search (NEMoTS), which is a novel combination of monte-carlo tree search and neural networks. Empirical results demonstrate the the method is adept at finding symbolic representations for time series while also being considerably faster than previous approaches.

**Strengths:**

I want to preface this by saying that I am not an expert in this field. In general, the paper is very well-written! It motivates the problem extremely well and at, a high level, the general principle easy to understand!

**Weaknesses:**

I again want to preface this by saying that I am not expert, though I hope a non-expert's viewpoint can help strengthen the paper. While I think the writing is great, everything is too surface level and there is not enough detail to reimplement this. As an example, in section 3.2.1, the authors state "The selection phase commences at the ”Root” node and involves iterative selection of child nodes (potential mathematical operations from the pre-defined function library) until a partially expanded or unexpanded node is encountered". What does a partially expanded or unexpanded node mean in this setting? Moreover, what is a leaf node? Is it an expression in the function library?

Next, the simulation section is also confusing to me. What exactly is being simulated? Does the simulation correspond adding more random functions?

It is also not clear how multiple datasets are leveraged here. A priori, I would expect each dataset corresponds to a different an expression, which would require its own policy and value network.

Lastly, there doesn't seem to be any details for reproducing the experimental results. What were the equations in the pre-defined library? What were the hyper-parameters of the MCTS? What were the hyper-parameters of the neural network? Without these details, it is impossible to reproduce the results of this paper.

**Questions:**

All my questions are in the weakness section.

---

### Official Review · Reviewer_XyLS · 2024-11-04

**Soundness:** 2
**Presentation:** 2
**Contribution:** 2
**Rating:** 5
**Confidence:** 3

**Summary:**

The paper proposes a new symbolic regression algorithm based on Monte Carlo Tree Search (MCTS) and evaluates it on the problem of time series analysis. It enhances the traditional MCTS with neural networks, forming Neural-Enhanced Monte-Carlo Tree Search (NEMoTS).

**Strengths:**

- The problem of fitting a symbolic expression to time series data is well-motivated.
- Different elements of the model are thoroughly described.
- The method achieves superior performance on the tested datasets compared to a few baselines.

**Weaknesses:**

- Positioning with respect to related works could be improved. Although the paper mentions symbolic regression methods based on MCTS by Kamienny et al. (2023) and Sun et al. (2022) it does not discuss works by Lu et al. (2021), Shojaee et al. (2023), and Xu et al. (2024). It is important to explain well the novelty compared to similar solutions.
- The problem setting deviates from the standard symbolic regression setting. Usually, symbolic regression methods are applied to general regression settings, whereas here it is constrained to a single univariate sample. I wonder why the authors decided to focus on this setting in particular instead of evaluating it on more standard settings and datasets.
-  Although the paper provides a comparison to another MCTS-based SR model by Sun et al. (2022), there are other MCTS-based methods and the experimental section would benefit by comparing with them as well.
- The results are not accompanied by any error bars, so it is difficult to gauge how significant the differences are or how robust the method is.
- I do not understand why extrapolation performance results in Table 2 do not include the symbolic regression methods used earlier.
- There is no measure of how interpretable the equations found by NEMoTS (or other methods) are.

**References**

Kamienny, P. A., Lample, G., Lamprier, S., & Virgolin, M. (2023, July). Deep generative symbolic regression with Monte-Carlo-tree-search. In International Conference on Machine Learning (pp. 15655-15668). PMLR.

Sun, F., Liu, Y., Wang, J. X., & Sun, H. (2022). Symbolic physics learner: Discovering governing equations via monte carlo tree search. arXiv preprint arXiv:2205.13134.

Lu, Q., Tao, F., Zhou, S., & Wang, Z. (2021). Incorporating Actor-Critic in Monte Carlo tree search for symbolic regression. Neural Computing and Applications, 33, 8495-8511.

Shojaee, P., Meidani, K., Barati Farimani, A., & Reddy, C. (2023). Transformer-based planning for symbolic regression. Advances in Neural Information Processing Systems, 36, 45907-45919.

Xu, Y., Liu, Y., & Sun, H. (2024). Reinforcement Symbolic Regression Machine. In The Twelfth International Conference on Learning Representations.

**Questions:**

1. What are the main novel elements compared to other SR methods based on MCTS?
2. Why did you decide to focus on time series analysis instead of a more general symbolic regression setting?
3. Why didn't you include symbolic regression methods in Table 2?

---

### Note · Authors · 2024-11-24

I have read and agree with the venue's withdrawal policy on behalf of myself and my co-authors.